# Treatment of Condyloma Acuminata with Tirbanibulin 1% Ointment in People Living with HIV: A Case Series with Literature Review

**DOI:** 10.3390/idr17030040

**Published:** 2025-04-25

**Authors:** Fabio Artosi, Terenzio Cosio, Lorenzo Ansaldo, Alessandro Cavasio, Loredana Sarmati, Luca Bianchi, Elena Campione

**Affiliations:** 1Dermatology Unit, Department of Systems Medicine, University of Rome Tor Vergata, 00133 Rome, Italy; fabio.artosi@alumni.uniroma2.eu (F.A.); luca.bianchi@uniroma2.it (L.B.); 2Department of Laboratory Sciences and Haematological Sciences, Fondazione Policlinico Universitario “A. Gemelli” Istituto Di Ricovero e Cura a Carattere Scientifico (IRCCS), Università Cattolica Del Sacro Cuore, 00168 Rome, Italy; 3Infectious Diseases Unit, Department of Public Health and Infectious Diseases, Santa Maria Goretti Hospital, Sapienza University of Rome, 00185 Roma, Italy; lorenzo.ansaldo@alumni.uniroma2.eu; 4Clinical Infectious Diseases, Department of System Medicine, Tor Vergata University, 00133 Rome, Italy; rosarioalessandro.cavasio@ptvonline.it (A.C.); srmldn00@uniroma2.it (L.S.)

**Keywords:** papillomavirus, condyloma, HIV, tirbanibulin, field of cancerization

## Abstract

Background: Condyloma acuminata (CA) are dysplastic lesions caused by human papillomavirus (HPV) infection. Condylomata acuminata are common in Human Immunodeficiency Virus- (HIV) infected individuals and have been linked to HIV transmission. Current therapeutic options for CA encompass laser, cryotherapy, imiquimod, sinecatechins, podophyllotoxin, and trichloroacetate. These topical therapies have limitations caused by significant local skin reactions, high recurrence rates, prolonged application times, and, in some cases, a supposed lower efficacy in people living with Human Immunodeficiency Virus (PLWH). Previous studies evaluated the effect in the CA treatment of tirbanibulin 1% ointment since it is a synthetic antiproliferative drug approved for the topical treatment of actinic keratoses, acting in two distinct ways: it inhibits microtubule polymerization and Src kinase signaling. Human papilloma virus can up-regulate the kinases Src and Yes, so the tirbanibulin efficient treatment of CA may be due to the suppression of Src kinase signaling. Methods: Here, we present for the first time a retrospective case series of three PLWHIV affected by CA. Case: The patients experienced variable outcomes, with complete resolution of smaller condylomas for 2 out of 3 patients. Adverse events were local and of mild to moderate severity, lasting one week or less. Conclusions: While in need of larger studies, it is possible to hypothesize tirbanibulin 1% ointment as a therapeutic alternative for people living with HIV, especially for condylomas smaller than 1 cm in size.

## 1. Introduction

Human papillomavirus (HPV) is a double-stranded circular DNA virus belonging to the papillomavirus family. It is transmitted by skin-to-skin or mucosa-to-mucosa contact and enters the body via cutaneous or mucosal trauma [1]. Human papillomavirus infection is one of the most common sexually transmitted diseases in humans; most sexually active people have at least one strain of HPV in their lifetime [2,3]. Of the more than 100 known strains of HPV, 13 are considered high risk (cancer-causing). Low-risk HPV strains 6 and 11 are linked to the development of condylomata acuminata (CA), also known as anogenital warts, which are benign epithelial lesions that frequently appear at locations prone to abrasion or injury during sexual intercourse [4,5]. Condylomata acuminata are common in Human Immunodeficiency Virus (HIV)-infected individuals and sexually active groups at risk of HIV acquisition and have been linked to HIV transmission [6]. Interestingly, new evidence seems to link HPV clearance, rather than HPV infection, more closely to HIV acquisition, contrary to those imagined. There remains, however, an intimate link between the two infections [7,8]. Hand to hand with local treatment, the goal after the discovery of an HPV infection is the eradication of high-risk (hr) HPVs and low-risk HPV infections, which is also the rationale for the vaccine [9]. Physical and pharmaceutical therapies for CA are currently available, including laser, cryotherapy, imiquimod, sinecatechins, podophyllotoxin, and trichloroacetate [10]. Still, a formal treatment algorithm does not exist, and treatment depends on lesion location, morphology, and patient preference [10,11,12]. These topical therapies have limitations caused by significant local skin responses (LSRs), high recurrence rates, or restricted accessibility and prolonged application times [13]. Tirbanibulin 1% ointment is a synthetic antiproliferative drug licensed by the Food and Drug Administration in 2020 for the once-daily topical treatment of actinic keratoses for 5 days. Tirbanibulin acts in two distinct ways: it inhibits microtubule polymerization by binding tubulin, and it inhibits Src kinase signaling in actively dividing cells [14,15,16]. Previous studies have evaluated the effect of tirbanibulin in the CA treatment and demonstrated complete resolution of recalcitrant genital CA and HPV-16 vulvar high-grade squamous intraepithelial lesions [17,18,19,20]. To this date, there are no recognized cases of CA treated with tirbanibulin in PLWHIV.

## 2. Material and Methods

### 2.1. Search Strategy

We performed a comprehensive search in the following databases from 2011 to January 2025: Cochrane Central Register of Controlled Trials; MEDLINE; Embase; US National Institutes of Health Ongoing Trials Register; NIHR Clinical Research Network Portfolio Database; and the World Health Organization International Clinical Trials Registry Platform. We studied reference lists and published systematic review articles. We used the term “tirbanibulin” or “KX2-391” with the following keywords, in combination: “papillomavirus”, “HPV”, “Condyloma acuminata”, “wart”, and “anogenital wart”. Only English language articles were included in the searches. Forward citation searching of reference lists of the original studies and review articles was also conducted.

### 2.2. Inclusion Criteria

To investigate the use of tirbanibulin in the treatment of anogenital warts, if the study included tirbanibulin with other drugs, only the tirbanibulin frame was analyzed. All human studies were included, with no restrictions on age, sex, ethnicity, or the type of study. Case reports and case series were included if they described papillomavirus infections treated with tirbanibulin.

### 2.3. Exclusion Criteria

The target intervention excluded the analyses of other pathologies that are not anogenital warts, and non-English language articles.

## 3. Results

Six articles or trials regarding the use of tirbanibulin in anogenital warts were identified by this quantitative research. One was excluded after the application of exclusion criteria. Among the five articles or trials eligible for evaluation, one was excluded after abstract or full text reading. Three articles or trials were evaluated in this case report (Figure 1).

In Table 1, there is a summary of evidence published to date (according to the search criteria) regarding the treatment of CA with tirbanibulin 1% ointment, including the baseline characteristics and number of patients treated, adverse events, and clinical outcomes observed.

In this case series, we described the first documented cases of CA treated with tirbanibulin 1% ointment and in PLWHIV.

## 4. Case Series

This is a retrospective case series of 3 PLWH treated with off-label tirbanibulin, with 1% ointment for CA. The patient demographic and clinical features are shown in Table 2.

Patients were visited on the day they started treatment with tirbanibulin 1% ointment (W0), after 1 week (W1) and after 10 weeks from the start of treatment (W10). During each visit, patients completed the EQ-5D-5L questionnaire and clinical photos of the affected area were taken. At the baseline visit (W0), patients were instructed on how to apply 1% tirbanibulin ointment (after cleansing the area, a layer of ointment is applied with circular movements until completely absorbed only on the lesions, once daily for five days). Local tolerability was measured at W0 and W10 by the EQ-5D-5L questionnaire, a patient-reported outcome tool introduced by the EuroQol Group in 2009 [21]. It is divided into two sections: a five-item questionnaire (Q) about mobility, self-care, usual activities, pain/discomfort, and anxiety/depression, and a second section with a Visual Analogue Scale (VAS) about a patient’s perception of their health. Its values can be reported as either disaggregated by domain or as a single summary index encompassing all five dimensions. We adopted the value set for Italy proposed by Finch et al. [22], according to which the “Q” part of the EQ-5D-5L yields a score ranging from 0 to 1.571, with higher values indicating greater impairment in the dimensions assessed by the questionnaire. The VAS part of the EQ-5D-5L questionnaire is a self-rating question of overall health status, measured on a 0–100 visual analogue scale [21].

Patient III was a candidate for retreatment with tirbanibulin 1% ointment at W10; therefore, he was scheduled for a further follow-up visit 16 weeks after the start of treatment. The study was conducted following the ICH/GCP guidelines and approved by the Institutional Review Board’s (IRB) Independent Ethical Committee Tor Vergata University Hospital (2.25CS; approved on 16 January 2025). Written informed consent was obtained from the patients to publish this case series and any accompanying images.

### 4.1. Patient I

A 37-year-old male, Caucasian, with borderline personality disorder, no other noteworthy comorbidities. HIV infection was diagnosed in 2012, with CD4^+^ nadir of 398 cells/mmc, zenith HIV-RNA 206,537 cp/mL (CDC stage A2). Over the years, the patient has had Highly Active AntiRetroviral Therapy (HAART) adherence issues, reporting voluntary discontinuations of several indicated regimens during follow-up visits. The patient is currently on regular HAART with DRV/c/TAF/FTC, and three-dose nonavalent-HPV vaccination with Gardasil9^®^ was completed in 2023. During the sexually transmitted infections (STIs) screening carried out annually, the anogenital physical examination revealed the presence of CA in the perianal area. Anorectal swabs were carried out to search for HPV-DNA, with confirmation of high-risk genotype 52 and low-risk genotype 6, 40 and 43, in association with an anal PAP test showing atypical squamous cells of undetermined significance (ASC-US) according to the Bethesda System [23]. Therefore, the patient underwent high-resolution anoscopy (HRA) to search for suspicious lesions on which to perform a possible biopsy, as per the European AIDS Clinical Society (EACS) guidelines [24]. Histological examination of the anal mucosa was positive for anal intraepithelial neoplasia—1 according to the Bethesda System [23].

The patient underwent a dermatological examination for the diagnosis of perianal condylomatosis. During the dermatological visit, four extra-rectal condylomatous lesions are appreciated and scattered over a rather large area (Figure 2A). The patient reports having undergone five previous sessions of cryotherapy with liquid nitrogen for perianal AC with little or no efficacy. We decided to prescribe treatment with tirbanibulin 1% ointment once a day for 5 consecutive days. The patient filled out the EQ-5D-5L questionnaire (Table 3). During a follow-up visit, at W2, the patient reported burning and moderate pain that began on the fourth day of application of the drug and lasted for approximately 7 days. On a dermatological physical examination, mild erythema and oedema of the treated area can be appreciated (Figure 2B). Pre-existing condylomatous lesions remain. At the next visit (W10), the patient showed a total *restitutio ad integrum* of the perianal skin and mucosa with the persistence of the CA present at baseline (Figure 2C). The patient fills out the EQ-5D-5L questionnaire again (Table 3). It was decided not to carry out retreatment with tirbanibulin, and cryotherapy was performed on the three main lesions. The first part (Q) of the EQ-5D-5L questionnaire showed a significant increase in the score from 0.249 at baseline to 0.833 at W1 (Table 3). This increase reflects a worsening in mobility, pain, and discomfort, likely associated with the side effects of the treatment in the days immediately following its completion. In contrast, the VAS component of the EQ-5D-5L remained relatively stable throughout the follow-up visits.

### 4.2. Patient II

A 47-year-old Caucasian male, previous occasional use of inhaled narcotics, no noteworthy pathology in the clinical history. Diagnosis of HIV infection was made in 2018, in association with the finding of chronic active Hepatitis B virus (HBV) infection. At diagnosis, the patient was classified as CDC stage B3, with CD4^+^ nadir of 193 cells/mmc, HIV-RNA zenith of 447,000 cp/mL. During the follow-up, the patient voluntarily interrupted HAART therapy for approximately three months, which has currently been resumed with BIC/TAF/FTC, taken regularly and in the absence of AEs. At the last viro-immunological check-up, the CD4^+^ cell count was 529 cells/mm, HIV-RNA 34 cp/mL, and CD4^+^/CD8^+^ ratio 1.78.

For evidence of perianal condylomatosis in the context of periodic visits, he underwent an anorectal swab to search for HPV DNA with a positive result due to the presence of high-risk (16, 31, 35, 53, 58, 73) and low-risk (6, 11, 42, 43, 54) genotypes. The patient has not undergone HPV vaccination and was included in the list for the anal PAP test.

The patient underwent a dermatological examination for suspected CA in the perianal area. At the dermatological objective examination, it was possible to diagnose perianal condylomatosis, with multiple exophytic lesions for which no pharmacological or physical therapy was ever administered (Figure 2D). The patient reports an unspecified allergy to local anaesthetic. The patient fills out the EQ-5D-5L questionnaire, the results of which are expressed in Table 2. A daily application of tirbanibulin 1% ointment in the perianal region is prescribed for 5 days. During W2, the patient reported burning and pain in the area of application of the drug, which began on the fifth day of application and lasted for approximately 4 days and was severe only on the first day of onset with slight difficulties in sitting, subsequently rapidly improving. Erythema and oedema of the treated area are appreciated (Figure 2E). At W10, the patient is re-evaluated and a reduction in the number of lesions compared to baseline is appreciated, especially of the smaller ones (Figure 2F). The patient fills out the EQ-5D-5L questionnaire again (results in Table 3). In agreement with the patient, it is decided to repeat the treatment with tirbanibulin 1% ointment in double administration: one sachet for the upper area and another for the lower area to be treated. The patient comes to the follow-up visit after 6 weeks (16 weeks since W0), presenting a substantial stability compared to the previous visit and a partial remission compared to the baseline (Figure 2G). The patient reports local AEs of moderate entity, complaining of mild itching and burning in the application area, milder than shown at W2. The patient was sent for a proctological surgical visit. Both the “Q” and the VAS part of the EQ-5D-5L questionnaire showed only minor modifications at the different follow-up visits, since the patient’s treatment-related discomfort lasted only 4 days.

### 4.3. Patient III

A 39-year-old Caucasian female patient diagnosed with mild hepatic steatosis in 2018, no other pathologies or surgical interventions reported in medical history. Also in 2018, following anaemia that was not well controlled by martial therapy and general malaise with significant asthenia, weight loss, and deterioration of her general conditions, on the advice of her general practitioner, she carried out HIV 1–2 Ag/Ab tests with a positive result. The patient was classified as B3 by CDC stadiation, with a CD4^+^ cell nadir of 75/mmc and a HIV RNA zenith of 108,000 copies/mL at baseline. Since diagnosis, follow-ups have been regular over the years, and the patient has continued to take ART with adherence, but showing only partial immunological recovery. The patient is currently taking BIC/TAF/FTC on a regular basis and without side effects. At the last viro-immunological check-up, the patient showed a CD4 cell count of 197/mmc, with undetectable HIV viremia. At the same time, cervico-vaginal swabs were carried out for HPV research with the presence of genotype 6, and anorectal swabs with detection of genotypes 6, 42, 70. The anal PAP test was negative.

The patient has been suffering from CA since 2016. She reports previous episodes of systemic allergy to penicillin and erythromycin. The patient reports allergies to several antibiotics for which she provides no further documentation. She underwent surgical excision of some perianal and rectal warts in 2017. Subsequently, she has undergone at least five previous cryotherapy interventions with liquid nitrogen for the treatment of perianal CA. At W0, the patient has at least five CA arranged radially around the anal orifice, with different dimensions (Figure 2G). At W0, the score of the EQ-5D-5L questionnaire is reported in Table 3. The patient returns for the W1 visit and complains of a moderately intense itching and burning sensation that, however, does not compromise her ability to carry out daily activities. The dermatological objective examination shows a partial de-epithelialization of the area site of the application of tirbanibulin with slight redness and oedema (Figure 2H). An emollient cream based on vitamin E is prescribed and a follow-up visit is scheduled. During the W10 visit, the patient fills out the EQ-5D-5L questionnaire, the scores of which are shown in Table 3. The patient reports the disappearance of symptoms approximately 7 days after applying tirbanibulin ointment. An inspection of the anogenital area shows a partial response, with resolution of the lesions localized on the left perianal margin, with permanence of the larger (~1 cm) right lesion (Figure 2I,J). The patient underwent cryotherapy only for the residual lesion. Evaluation of the EQ-5D-5L questionnaire revealed no changes in the score of the first section, as the patient reported no difficulties in performing normal movements or activities of daily living during the follow-up visits. Regarding the VAS component, the patient reported a lower score of 50 at W10, compared to 65 at baseline, likely reflecting the psychological impact of the cryotherapy treatment required due to a partial clinical response to tirbanibulin ointment.

## 5. Discussion

Immunocompromised and HIV-positive patients carry a high epidemiological burden of anogenital human papillomavirus-associated lesions, and it is often difficult to clinically differentiate CA from intraepithelial neoplasia [23]. In the therapeutic armamentarium for CA, we include physical, chemical, and pharmacological methods, as laser, cryotherapy, imiquimod, sinecatechins, podophyllotoxin, and trichloroacetate, burdened by predominantly local adverse events (AEs). None of these are suitable for all patients and the same severity of disease. The recurrence rates for CA treated with topical podophyllotoxin treatments have a clearance rate of 45 to 83 percent and a recurrence rate of 6 to 100 percent. Imiquimod treatments have a lower recurrence rate of 6 to 26 percent and a clearance rate of 35 to 68 percent. Sinecatechins ointment clears 47 to 59 percent of CA, and only 7 to 11 percent of patients have a recurrence of lesions after clearance. Cryotherapy with liquid nitrogen clears 44 to 75 percent of CA with a 21 to 42 percent recurrence rate. Treatment with trichloroacetic acid solution clears 56 to 81 percent of lesions, and 36 percent of lesions recur after clearance [25]. In 2019, Barton et al. [26] demonstrated that carbon dioxide laser therapy is the most effective treatment for achieving a complete clearance of CA at the end of treatment. Of patient-applied topical treatments, podophyllotoxin 0.5% solution was found to be the most effective at achieving complete clearance and was associated with a statistically significant difference compared with imiquimod 5% cream and polyphenon E 10% ointment [26,27]. Moreover, imiquimod and podophyllotoxin creams had similar efficacy for CA clearance. Still, the results do not support earlier evidence of a lower recurrence with the use of imiquimod than with podophyllotoxin [28]. Despite the available armamentarium, few studies regarding the efficacy of those molecules in people living with HIV have been published. Bilenchi et al. [29] demonstrated the efficacy of sinecatechins 10% ointment against CA in a 55-year-old woman living with HIV, after the failure of a combination of cryotherapy and imiquimod 5% cream. Saiag et al. [30] demonstrated that Imiquimod 5% cream did not show safety concerns and is suitable for use in PLHIV with CA and successful HAART treatment, but total CA clearance was observed in 32% and the HPV load decreased or became undetectable in 40% of the patients at week 16. Although a known reason is lacking, it seems that PLWHIV responds worse to topical therapy with imiquimod 5% cream for CA than non-HIV patients. This finding adds to the daily therapeutic challenge that clinicians face when treating patients affected by CA with the complex immunological asset given by HIV co-infection [31]. The search for safe, fast, and effective therapies in HPV treatment should also consider the patient’s perspective and the disease-related stigma that often stems from societal judgments about the presumed behaviors that “caused” the condition [32]. Previous research has consistently reported that the fact that HPV is sexually transmitted significantly increases internalised stigma and, consequently, feelings of shame [33].

HPV 16 has been extensively studied for its ability to up-regulate the Src family kinases (SFKs) and Yes via posttranscriptional pathways [34]. SFKs are intracellular nonreceptor tyrosine kinases that participate in various signaling cascades. It is now known that SFKs play an essential role in human epithelium malignancies and have been linked to the growth of colon and breast cancers. E6 and E7 HPV-oncoproteins inactivate p53 and Rb proteins, resulting in uncontrolled proliferation. Szalmas et al. [35] found that HPV 16 E6 and E7 increased Src and Yes expression levels in human keratinocytes and increased the activating phosphorylation of all accessible SFKs. Tirbanibulin inhibits tubulin signaling by acting on a microtubule blockade and phosphorylating Src. The signal transduced by Src needs intracellular microtubule activity, so tirbanibulin also inhibits this group of proteins in several pathways. These data should suggest that tirbanibulin could be an efficient treatment against CA, maybe due to the suppression of Src kinase signaling [15,16], despite further in vitro and in vivo studies being necessary to evaluate the impact of tirbanibulin on low-risk HPV etiopathogenesis in CA. Tirbanibulin has shown efficacy as a cancer field therapy against actinic keratoses [36], promoting its action on several lesions simultaneously, limiting the certain occurrence of local AEs associated with physical therapy (e.g., liquid nitrogen cryotherapy) on multiple lesions in the same session. Similarly, we believe it is reasonable to hypothesize a similar action for tirbanibulin in settings where multiple condylomata are present, for which cryotherapy over an extended treatment field is not readily tolerated because of the frequent occurrence of moderate-grade local AEs [37]. This hypothesis is also reinforced by the efficacy of two cycles of 15 days of topical 1% tirbanibulin ointment for a plantar viral wart, another skin lesion given by HPV infection [38]. In light of our results, tirbanibulin 1% ointment would appear to be most effective on small lesions, smaller than about 1 cm, probably of a more recent onset, with moderate local AEs lasting about 1 week. The duration of local AEs was shorter than previously reported by the other authors [18], and when the treatment was repeated, local AEs were milder than in the first control. This may lead clinicians to consider retreatment, especially for small lesions.

The data on patient-reported outcomes including QoL are still lacking, which is an essential limitation since patient-centred outcomes should play an important role when evaluating treatment strategies. However, apart from the available evidence, treatment choice also depends on individual parameters and patient preferences. For patient I, Part 1 of the EQ-5D-5L questionnaire recorded an increased score at W1 due to the pain and burning experienced at the perianal site, which affected the regular conduction of daily activities and intimate hygiene. In light of the score results, patients considered themselves to be in good health on average. Patient III, despite experiencing a reduction in the number of CA, reported a perceived decline in health status on the VAS of the EQ-5D-5L at W10. This was likely due to the psychological burden of requiring additional cryotherapy in addition to previous local discomfort for topical tirbanibulin ointment application, which she interpreted as an indication of incomplete healing. This might lead us to think that adequate regimens of an effective topical drug reduce the psychological burden of AC patients facing multiple sequential therapies for the clearance of perianal lesions.

## 6. Conclusions

Management of HPV infection is a challenge in clinical practice due to significant LSRs, high recurrence rates, or restricted accessibility and prolonged application times of actual treatments. Condyloma acuminata management must be customized to improve the patient’s QoL hand-to-hand with clinical and biological cures. Our study demonstrated that the local application of tirbanibulin 1% ointment was efficacy on CA smaller than 1 cm, with mild to moderate local AEs during about 7 days in PLHIV.

Despite our clinical experience, the inclusion of only three patients precludes any robust statistical analysis. Therefore, future studies involving larger cohorts are necessary to validate these preliminary findings, as well as in vitro experiments to elucidate the impact of low-risk HPV genotypes on the expression of SFKs.

## Figures and Tables

**Figure 1 idr-17-00040-f001:**
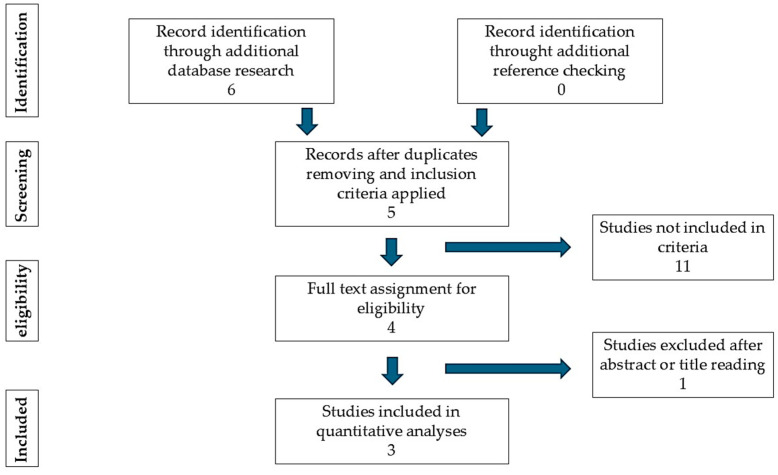
Flow chart reporting the data selection and analysis method used.

**Figure 2 idr-17-00040-f002:**
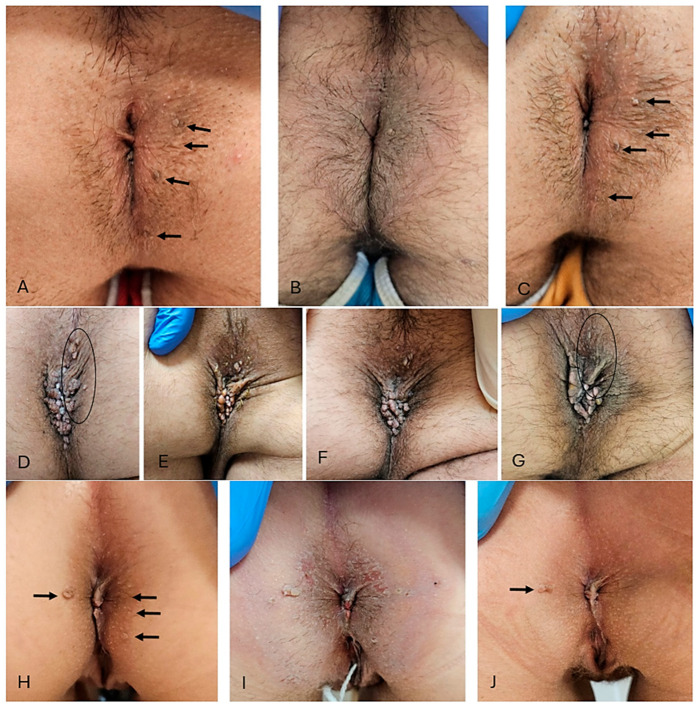
Clinical presentation and follow-up of CA treated with tirbanibulin 1% ointment. Patient I. Perianal CA pretreatment (W0; black arrows; (**A**)); post-treatment with the first (W1; black arrows; (**B**)) and follow-up (W10; black arrows; (**C**)). Patient II. Perianal CA pretreatment (W0; black arrows; (**D**)); post-treatment with the first (W1; black arrows; (**E**)), second (W8; black arrows; (**F**)) and third (W12; black arrows; (**G**)) cycle. Patient III. Perianal CA pretreatment (W0; black arrows; (**H**)); post-treatment with the first (W1; black arrows; (**I**)) and follow-up (W10; black arrows; (**J**)). CA, condyloma acuminata.

**Table 1 idr-17-00040-t001:** Tirbanibulin usage in CA treatment.

	Characteristics of Population	Number of Patients	Adverse Events	Outcome
Braasch et al., 2022 [18]	Male with recurrent CA	2	Mild local inflammation, redness, heat sensation	Partial resolution after 1 or 2 cycles of application
Moore et al., 2023 [19]	3 male and 2 female with recurrent CA	5	Local reactions in 3 patients from mild to severe. Reported symptoms: erythema, scaling, crusting, dryness, and post-inflammatory depigmentation. Complete resolution in 21 days.	Complete resolution after 1 to 5 rounds (mean 2)
Agostini et al., 2024 [20]	1 male and 1 female with no previous treatments in the treated area	2	Mild to moderate local inflammatory reaction, well tolerated by the patients.	Complete resolution after 60 days since treatment and 1 round of treatment

**Table 2 idr-17-00040-t002:** Clinical characteristics of patients treated with tirbanibulin 1% ointment. AE = Adverse Effects; AIN-1 = Anal Intraepithelial Neoplasia 1 according to Bethesda System; CA = Condyloma Acuminata; MSM = Men who have sex with men; NA = Not available; 9vHPV = Nonavalent HPV vaccine (Gardasil9^®^).

	Patient I	Patient II	Patient III
Age	37	47	39
Sex	Male	Male	Female
Ethnicity	Caucasian	Caucasian	Caucasian
Infection route	Sexual	Sexual	Sexual
Sexual behavior	MSM	Heterosexual	Heterosexual
Years since HIV diagnosis	11	4	6
CD4^+^ nadir (cells/mmc)	398	193	75
HIV-RNA zenith (cp/mL)	206,537	447,000	108,000
CDC Stage	A2	B3	B3
Currently on HAART	Yes	Yes	Yes
Current HAART	DRV/c/TAF/FTC	BIC/TAF/FTC	BIC/TAF/FTC
Last CD4^+^ count (cells/mmc)	329	529	197
Last HIV-RNA (cells/mmc)	104	34	TND
Last CD4^+^/CD8^+^	0.31	1.78	0.31
HPV genotypes isolated	6, 40, 43, 52	6, 11, 16, 31, 35, 42, 43, 53, 54, 58, 73	6, 42, 70
HPV vaccination	9vHPV	no	9vHPV
CA localizations	Perianal	Perianal, rectal	Perianal
CA number	4	Multiple	4
CA dimension	0.8–1 cm	Clustered lesions > 1 cm; single speckled lesions < 1 cm	1.5 cm (left lesion); 0.5–0.7 cm (right lesions)
Other HPV-related lesions	AIN-1, p16 weakly positive	NA	Anal L-SIL in 2019
Previous CA treatments	Five previous cryotherapy sessions	No previous treatment for CA	Surgery and more than 5 previous cryotherapies
CA Clinical resolution	No	Partial resolution of single speckled lesions < 1 cm	Complete resolution for right lesions; no response for left lesion
N. of rounds	1	1 + (1 + 1)	1
AEs reported	Burning and pain	Burning, pain, and difficulty in sitting for approximately 3 days after the end of treatment	Burning and pain

**Table 3 idr-17-00040-t003:** EQ-5D-5L questionnaire results.

	Patient I		Patient II		Patient III	
EQ-5D-5L	Q	VAS	Q	VAS	Q	VAS
Baseline	0.249	75	0.369	70	0.045	65
W1	0.833	70	0.596	70	0.045	70
W10	0.334	70	0.369	70	0.045	50

Q: part 1 of questionnaire EQ-5D-5L; VAS: visual analogue scale part of questionnaire EQ-5D-5L.

## Data Availability

All data generated or analyzed during this study are included in this article. Further inquiries can be directed at the corresponding author.

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
