# Peer review of "Treatment of Condyloma Acuminata with Tirbanibulin 1% Ointment in People Living with HIV: A Case Series with Literature Review"

_2036-7449, 2025, doi:10.3390/idr17030040_

Round 1
Reviewer 1 Report
Comments and Suggestions for Authors
Page 2. Line 68.
Instead of the citation # 14 I would suggest to use these references:
Niu L, Yang J, Yan W, Yu Y, Zheng Y, Ye H, Chen Q, Chen L. Reversible binding of the anticancer drug KXO1 (tirbanibulin) to the colchicine-binding site of β-tubulin explains KXO1's low clinical toxicity. J Biol Chem. 2019 Nov 29;294(48):18099-18108
DeTemple VK, Walter A, Bredemeier S, Gutzmer R, Schaper-Gerhardt K. Anti-tumor effects of tirbanibulin in squamous cell carcinoma cells are mediated via disruption of tubulin-polymerization. Arch Dermatol Res. 2024 Jun 7;316(7):341.
Author Response
Reviewer 1
Thank you very much for allowing us to revise our manuscript (Submission ID: idr-3501259). We also would like to thank the reviewers for their valuable comments that undoubtedly have improved the quality of our paper.
Here, you will find a point-to-point letter in order to respond to the reviewer’s comments and suggestions. All the changes made are in red font (marked copy).
Page 2. Line 68. Instead of the citation # 14 I would suggest to use these references:
- Niu L, Yang J, Yan W, Yu Y, Zheng Y, Ye H, Chen Q, Chen L. Reversible binding of the anticancer drug KXO1 (tirbanibulin) to the colchicine-binding site of β-tubulin explains KXO1's low clinical toxicity. J Biol Chem. 2019 Nov 29;294(48):18099-1810
- DeTemple VK, Walter A, Bredemeier S, Gutzmer R, Schaper-Gerhardt K. Anti-tumor effects of tirbanibulin in squamous cell carcinoma cells are mediated via disruption of tubulin-polymerization. Arch Dermatol Res. 2024 Jun 7;316(7):341.
Thank you for your suggestion. We have modified the references according to your suggestions.
Finally, we have appreciated all of your feedback and have carefully considered your suggestions for improving our manuscript.
Reviewer 2 Report
Comments and Suggestions for Authors
The article reports the first cases of AC treatment among PLHIV using tirbanibulin. The authors should pay special attention to the presentation of tables and figures, as well as to the interpretation of the current Table 2, from which no results are mentioned or analyzed.
Specific comments are included in the attached file.

Author Response
Reviewer 2
Thank you very much for allowing us to revise our manuscript (Submission ID: idr-3501259). We also would like to thank the reviewers for their valuable comments, which undoubtedly improved the quality of our paper.
Here, you will find a point-to-point letter in order to respond to the reviewer’s comments and suggestions. All the changes made are in red font (marked copy).
- What do you mean by real cases? You can just write "cases."
- Thank you for your suggestion. We have modified the references according to your suggestions.
- This sentence is correct, but it should be deleted, as it suggests that AC can lead to anal cancer, which is incorrect. The study focuses on AC caused by low-risk HPV, so high-risk HPV should not be mentioned as often.
We have modified the sentence and outline this point in the discussion and in the conclusion sections.
- How was this review conducted? Was it systematic? More information is needed. This information, although included in supplementary material, should be added to the article in a paragraph detailing how the review was conducted, thereby avoiding the need for a supplementary document.
Dear reviewer, thank you for point this out. We have included the material and method section, as well as the results, in the main text and remove the supplementary files.
- All tables must have names. The explanation must be included in the text.
- It would be Table 2, Table 1 was the review. Review the numbering of tables and figures
- Check the order of the tables
- What was Figure A? Check the order of the figures or that they are all mentioned in the text.
Thank you for your suggestions. We have modified Figure and Tables according to your suggestions (list below).
Figure 1. Flow chart reporting the data selection and analysis method used.
Figure 2. Clinical presentation and follow-up of CA treated with tirbanibulin 1% ointment.
Table 1. Tirbanibulin usage in CA treatment.
Table 2. Clinical characteristics of patients treated with tirbanibulin 1% ointment.
Table 3. EQ-5D-5L questionnaire results.
- Authors should describe the results and interpret them, not just have the reader see the table.
- These results should be explained, if they are not important, they should not be mentioned.
- This table is mentioned several times, but at no point are the results or information explained.
Thank you for your suggestions. We have included and critically commented the results in Table 3.
- The table presentation could be improved so that it is on a single page and easier to analyze.
Thank you for your suggestions. We weren’t able to reduce the table in one page but we hope the editorial team could do that before publication.
- This isn't a route of infection. It's a classification of vulnerable groups and sexual preferences.
Thank you for your suggestions. We have modified the Table according to your suggestions, dividing route of infection and sexual behaviour.
- HPV-16 is a high-risk HPV, unlike the low-risk HPVs that cause AC, such as HPV-6 and HPV-8. These statements should be considered with caution, and the mechanism of action suggested by the authors should be considered as a hypothesis.
Thank you for your suggestions. We have modified the discussion section and report this limitation in the conclusion section.
- With 3 patients it is not possible to perform any statistical tests.
We have added this information in the study limitation section.
- This list of abbreviations is not required. All abbreviations must be mentioned first in the document.
Abbreviations’ list has been remove according to your suggestion.
Finally, we have appreciated all of your feedback and have carefully considered your suggestions for improving our manuscript.